# The relationship between N-terminal pro-brain natriuretic and coronary artery lesions in Kawasaki disease: A meta-analysis

Yanyan Li[ID][◉], Chaolong Zheng, Sisi Cheng, Limin Chu[◉], Zhiqing Chen, Guiling Liu*

Department of Pediatrics, The First Hospital of Hebei Medical University, Shijiazhuang, China

◉ These authors contributed equally to this work.
* 316853113@qq.com

## Abstract

### Background

Regarding the value of N-terminal brain natriuretic peptide precursors in evaluating coronary artery lesions (CAL) in Kawasaki disease.

### Objective

To conduct a meta-analysis of relevant literature and explore the value of N-terminal brain natriuretic peptide NT ProBNP in evaluating coronary artery lesions (CAL) in Kawasaki disease.

### Method

Retrieve publicly published literature on the relationship between serum NT ProBNP concentration and coronary artery disease in Kawasaki disease from CNKI, Wanfang, VIP, China Biomedical Database, Pubmed, Embase, Cochrane Library, Web of Science, and OVID databases, as well as manually search for other resource literature. Use state14 software to evaluate the quality of the included literature, calculate the combined sensitivity, specificity, positive likelihood ratio, negative likelihood ratio, and diagnostic ratio ratio, and compile a comprehensive subject working characteristic curve (SROC) to calculate the area under the curve (AUC) and 95% confidence interval.

### Result

A total of 25 articles were included using a random effects model, with a combined sensitivity of 0.80 [95% CI (0.74, 0.85)], specificity of 0.78 [95% CI (0.75, 0.82)], positive likelihood ratio of 3.94 [95% CI (3.32, 4.69)], negative likelihood ratio of 0.23 [95% CI (0.17, 0.30)], and diagnostic ratio of 17.24 [95% CI (12.0,24.77)]. The area under the SROC curve AUC is 0.87 (0.84–0.90). When the predicted value is

**Data availability statement:** All relevant data are within the manuscript and its Supporting Information files.

**Funding:** The author(s) received no specific funding for this work.

**Competing interests:** The authors have declared that no competing interests exist.

1000ng/L, the combined sensitivity is 0.87 [95% CI (0.80,0.92)], and the specificity is 0.78 [95% CI (0.71,0.84)]. Subgroup analysis revealed that the diagnostic value of NT ProBNP for Kawasaki disease complicated with CAL was higher in the over 3 year old group than in the under 3 year old group. NT ProBNP had higher diagnostic value for Chinese population, but its exclusion value was lower than that of Japanese population. The sensitivity and specificity of using chemiluminescence to detect NT-ProBNP were higher than those of enzyme-linked immunosorbent assay.

## Conclusion

NT-ProBNP has high diagnostic value for coronary artery lesions in Kawasaki disease, and has greater diagnostic value for children over 3 years old, and the use of chemiluminescence.It demonstrates superior diagnostic efficacy for the Chinese population and enhanced exclusionary power for the Japanese population.

## 1. Introduction

Kawasaki disease (KD), also known as cutaneous mucosal lymph node syndrome, is a systemic non-specific vasculitis that occurs in infants and young children, mainly involving small and medium-sized blood vessels. The main clinical symptoms are fever, mucositis, and rash. The incidence rate of Kawasaki disease (KD) in Southeast Asia has been increasing year by year for more than 20 years [1]. Early onset of Kawasaki disease is often characterized by fever, which may be accompanied by clinical manifestations such as cracked lips, hard swelling at the ends of hands and feet, and rash. Coronary artery disease (CAL)is the most serious complication. of Kawasaki disease, including coronary artery dilation, aneurysms, cardiomyopathy, etc, and is currently the main cause of secondary acquired heart disease in children.In particular, coronary artery disease is often detected 2–3 weeks after onset [2], Many children have developed coronary artery lesions at the time of diagnosis, posing great challenges to treatment. Moreover, the severity of CAL lesions and the intervention difficulty in the sequelae stage of Kawasaki disease are even higher than that of adult coronary heart disease [3].resulting in the decline of the quality of life of children, and bring serious family economic burden. Although echocardiography and coronary angiography are the gold standard for diagnosing coronary artery injury, many young children are difficult to soothe because they need complete quiet cooperation for echocardiography. Coronary angiography diagnosis requires interventional operation, which has great side effects and high cost for children, and many families of children have concerns [4]. At present, there is still a lack of early simple and accurate assessment of CAL in children with Kawasaki disease. Therefore, it is of great clinical significance to find indicators for early diagnosis of KD and prevention of CAL [5]. In recent years, some studies have confirmed that increased plasma NT-proBNP levels in the acute phase of KD is a high risk factor for coronary disease [6], and a significant positive correlation with coronary artery diameter, providing a basis for early diagnosis of coronary artery lesions in Kawasaki disease [7], but some studies

suggest that N-terminal brain sodium peptide precursor (NT-ProBNP) is meaningless or controversial [8,9] for predicting coronary artery lesions.Although the existing systematic evaluation [10] study NT-ProBNP evaluation kawasaki disease coronary artery lesions are valuable, but its included literature before 2020, and less literature, after 2020 recently published relevant literature, and the incidence of kawasaki disease in recent years, to update its diagnostic value, provide evidence-based basis for early clinical diagnosis of kawasaki disease and prevention of coronary artery lesions.

## 2. Materials and methods

### 2.1 Search strategy

CNKI, Wanfang, VIP, China biomedical database, Pubmed, embase, Cochrane Library, Web of Science, OVID, and included in the clinical case studies and column studies on the relationship between N-terminal brain natriuretic peptide precursor and cardiac injury in Kawasaki disease published from January 2014 to June 2025. The search scope includes full text of journals, master's and doctoral dissertations, guidelines, etc. Chinese search terms: N-terminal brain natriuretic peptide precursor, N-terminal brain natriuretic precursor,etc.; Kawasaki disease, mucocutaneous lymph node syndrome,etc.; coronary artery disease, coronary artery disease, etc. English search words: subject word Mucocutaneous lymphnode syndrome and its free words, subject word Coronary Artery Disease and its free words, subject word pro-brain natriuretic peptide and its free words.

### 2.2 Criteria for inclusion and exclusion

Inclusion criteria: 1. Published cohort study, medical record review, case-control study on the relationship between serum NT-ProBNP concentration and coronary artery lesions of Kawasaki disease, including typical Kawasaki disease and atypical Kawasaki disease; 2. Children under 14 years and infants; 3. Children were sampled before application of IVIG and aspirin, with clear criteria for diagnosis of Kawasaki disease and gold standard for clearly distinguishing coronary artery damage, coronary artery lesions and Kawasaki disease control group for non-coronary lesions; 3. Four table data of the diagnostic experiment. Exclusion criteria: 1. Repeated literature, 2, review, systematic evaluation, animal experiments and case reports, 3. the study content is not consistent, the control group is not the non-coronary disease group of Kawasaki disease, 4. insufficient data or repeated data, the study object is adults, 5. four tables cannot be extracted from the full text.

### 2.3 Data extraction and quality evaluation

**Data extraction.** The search results of different databases were imported into the literature management software Note Express. After the literature was included, two researchers evaluated the quality and quality of the literature. If there was any problem, the problem was negotiated with the third researcher. During literature screening, first read the literature title and abstract for preliminary screening. After excluding the literature that obviously does not meet the inclusion criteria, the remaining literature is determined to be included by reading the full text.

**Quality evaluation.** The studies included in the Meta analysis using the QUADAS-2 evaluation scale, the standard contains four parts, respectively case selection, evaluation of the experiment or interpretation, the gold standard implementation and interpretation of cases process, a total of 11 questions, according to the answer "yes", "no", "uncertainty", "corresponding to the risk of bias level can be judged as" low "," high "," "uncertainty". The specific contents are as follows: 1. Whether continuous or random medical records are included; 2. whether the medical record-control study design is avoided; 3. whether the study avoids inappropriate exclusion. 4. Is the interpretation of the results to be evaluated without knowing the results of the gold standard experiment? 5. If the threshold is used, is it predetermined? 6. Can the gold standard correctly distinguish between the target disease state? 7. Is the interpretation of the gold standard made without knowing the results of the selected test method? 8. Is there an appropriate time interval between

the evaluated test and the gold standard? 9. Did all the patients receive only one same criterion? 10. Did all patients receive only one same gold standard? 11. Are all the medical records included in the analysis?

### 2.4  Statistical analysis methods

Statistical analysis of the diagnostic four-grid tables extracted from the included literature was performed using state14 software. A heterogeneity test was performed first performed, and I2 was used to assess the sensitivity and the magnitude of specific heterogeneity of the included literature, I2 > 50%, indicating significant heterogeneity. Mmerge using random effects but not fixed effects. The combined sensitivity, specificity, positive likelihood ratio, negative likelihood ratio and diagnostic odds ratio, and the combined receiver operating characteristic curve (summary receiver operating characteristic curve, SROC) were calculated to calculate the area under the curve (area under the curve, AUC) and 95% confidence interval. The meta regression was used to find sources of heterogeneity and subgroup analysis was performed. Sensitivity analysis and publication bias test were performed using Stata14.0 software and Fagan plots were drawn to estimate post-test probability.

## 3.  Results

### 3.1  Literature screening results

A total of 512 articles were retrieved in this study. After eliminating duplicate documents, reading each abstract and full text, and screening strictly according to the inclusion and exclusion criteria, 25 articles were finally included, including 9 English articles including China, Japan, Russia and South Korea, and 16 articles in Chinese. In total, 2611 cases were included, including 648 in the coronary lesion group and 1963 in the non-coronary lesion group(NCAL), and the minimum number of included cases was 31 and 336. The literature screening process is shown in Fig 1, the basic information of the included literature is shown in Table 1, and the diagnostic parameters are shown in Table 2.

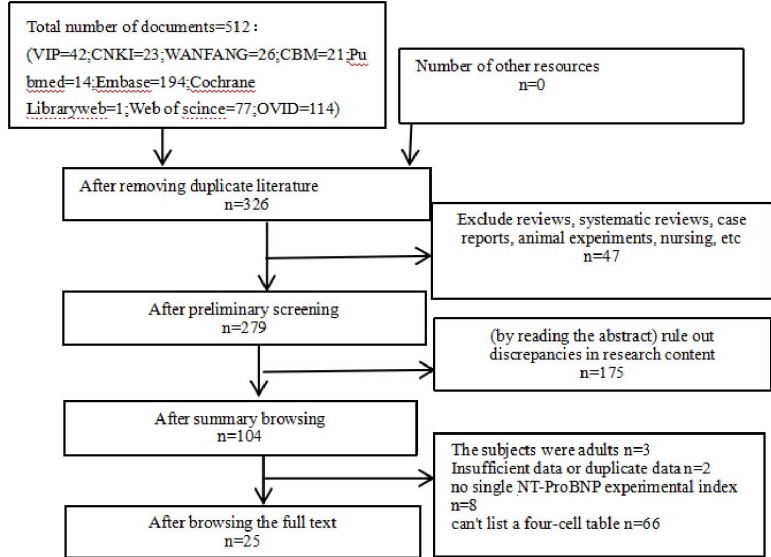

**Fig 1.  Literature screening flowchart.flow-diagram for study collection, showing number of studies identified, screened, eligible.**

**Table 1. Basic information of included literature.**

| Author | Year | Country | Research method | Is blood collected before treatment | CAL | NCAL | NT-ProBNP detection method | Median age |
|---|---|---|---|---|---|---|---|---|
| Jae Yun Jung [11] | 2018 | korea | Case Review | yes | 23 | 86 | —— | 2.7 |
| Hye Young Lee, MD [12] | 2016 | korea | Case control | yes | 30 | 257 | —— | 2.2 |
| S.Kurbanova1 [13] | 2022 | Russia | Case control | yes | 61 | 127 | —— | 2 |
| Kazunari Kaneko [14] | 2011 | Japan | Case control | yes | 6 | 37 | Chemiluminescence immunity | 2.17 |
| Ken Yoshimura [15] | 2012 | Japan | Case control | yes | 7 | 58 | —— | —— |
| Ken Yoshimura MD [16] | 2013 | Japan | Case control | yes | 19 | 61 | Chemiluminescence immunity | 1.7 |
| Kaoru Satoh [17] | 2017 | Japan | Case Review | yes | 7 | 24 | —— | 2.86 |
| Chou huixian [18] | 2012 | China | Case control | yes | 25 | 77 | —— | 1.67 |
| Chenyanli [19] | 2014 | China | Case control | yes | 9 | 26 | Enzyme linked immunosorbent assay | 1.7 |
| Lu Huiling [20] | 2014 | China | Case control | yes | 33 | 73 | Enzyme linked immunosorbent assay | 2.63 |
| Zeng, A [21] | 2017 | China | Case control | yes | 20 | 23 | —— | —— |
| Jiang yajing [22] | 2018 | China | Case control | yes | 17 | 33 | Chemiluminescence immunity | 3.2 |
| Guo yonghong [23] | 2019 | China | Case control | yes | 16 | 53 | Chemiluminescence immunity | 2.98 |
| Wang runbang [24] | 2020 | China | Case control | yes | 19 | 40 | Enzyme linked immunosorbent assay | 2.76 |
| Liu gengying [25] | 2021 | China | Case Review | yes | 57 | 279 | —— | —— |
| Xue qiuyu [26] | 2021 | China | Case control | yes | 17 | 67 | —— | 1.73 |
| Bing Bai [27] | 2022 | China | Case Review | yes | 31 | 40 | —— | 1.89 |
| Yang yanfeng [28] | 2022 | China | Prospective cohort study | yes | 45 | 195 | Immunoassay method | 2.71 |
| Lv aiting [29] | 2022 | China | Case control | yes | 29 | 59 | —— | 3.50 |
| Que xuejun [30] | 2022 | China | Case control | yes | 41 | 57 | Chemiluminescence immunity | 1.11 |
| Sun xinna [31] | 2022 | China | Case Review | yes | 29 | 51 | Chemiluminescence immunity | 3.12 |
| Jiao lihua [32] | 2023 | China | Case control | yes | 47 | 65 | Chemiluminescence immunity | 3.25 |
| Cao hong [33] | 2023 | China | Case control | yes | 19 | 80 | Immunoassay method | 1.75 |
| Dong nan [34] | 2023 | China | Case Review | yes | 11 | 53 | Enzyme linked immunosorbent assay | 4.25 |
| Liu ya [35] | 2023 | China | Case control | yes | 30 | 42 | Enzyme linked immunosorbent assay | 2.8 |

CAL: Coronary artery disease. NCAL: non-coronary lesion group."——": No relevant information is mentioned in the article.

### 3.2 Literature quality evaluation

The quality of the included literature was evaluated. Case selection: Continuous or randomized medical records were included in all literatures, and case-control study designs and inappropriate exclusions were avoided, indicating no bias in case selection; Implementation or interpretation of the experiments to be evaluated: There were 10 [11,13,15,17,21,23,27,31,34,35] literatures whose results of the trials to be evaluated were interpreted without knowing the results of the gold standard experiments. In 16 [13–19,21–24,28,31–34] literatures, threshold values were used but not determined first.There were 2 [34, 35] literatures that did not use the threshold value, suggesting that the implementation or interpretation of the experiment to be evaluated resulted in a large bias. Implementation and interpretation of the gold standard: All the gold standards selected in the study could correctly distinguish the target disease states, and there were 5 [11,12,14–16] articles in which the gold standard was interpreted with knowledge of the results of NT-ProBNP, indicating that the implementation and interpretation of the gold standard may produce a small bias. Procedure of cases: There were 3 [14,16,20] literatures with a long time interval between the trial to be evaluated and the gold standard, and 1 [31] literatures with no clear explanation. All patients selected in the literatures received the same standard and the gold standard. Moreover, all medical records were included in the analysis, which showed that basic bias occurred in the process of the

**Table 2. Included diagnostic parameters.**

| Author | Year | Parameter | Tp | Fp | Fn | Tn | Sp | Estimate(ng/L) |
|---|---|---|---|---|---|---|---|---|
| Jae Yun Jung [11] | 2018 | NT-ProBNP | 18 | 33 | 5 | 53 | 78.26% | 515.4 |
| Hye Young Lee, MD [12] | 2016 | NT-ProBNP | 29 | 9 | 4 | 64 | 89.00% | 950 |
| S.Kurbanova1 [13] | 2022 | NT-ProBNP | 53 | 30 | 8 | 95 | 87.50% | 1015 |
| Kazunari Kaneko [14] | 2011 | NT-ProBNP | 5 | 12 | 1 | 25 | 83% | 1000 |
| Ken Yoshimura [15] | 2012 | NT-ProBNP | 6 | 18 | 1 | 40 | 86% | 1000 |
| Ken Yoshimura MD [16] | 2013 | NT-ProBNP | 18 | 9 | 1 | 52 | 95% | 1300 |
| Kaoru Satoh [17] | 2017 | NT-ProBNP | 5 | 4 | 2 | 20 | 80% | 1555 |
| Chou huixian [18] | 2012 | NT-ProBNP | 21 | 23 | 4 | 54 | 84% | 827 |
| Chenyanli [19] | 2014 | NT-ProBNP | 7 | 7 | 2 | 19 | 78% | 900 |
| Lu Huiling [20] | 2014 | NT-ProBNP | 22 | 83 | 8 | 174 | 73.30% | 853.4 |
| Zeng, A [21] | 2017 | NT-ProBNP | 16 | 5 | 4 | 18 | 82.80% | 945.9 |
| Jiang yajing [22] | 2018 | NT-ProBNP | 15 | 3 | 2 | 30 | 89.21% | 1070 |
| Guo yonghong [23] | 2019 | NT-ProBNP | 13 | 11 | 3 | 42 | 81.30% | 756.32 |
| Wang runbang [24] | 2020 | NT-ProBNP | 16 | 11 | 3 | 29 | 84.20% | 463.75 |
| Liu gengying [25] | 2021 | NT-ProBNP | 52 | 62 | 5 | 217 | 91.80% | 382.5 |
| Xue qiuyu [26] | 2021 | NT-ProBNP | 17 | 21 | 0 | 46 | 100% | 595.1 |
| Bing Bai [27] | 2022 | NT-ProBNP | 16 | 3 | 15 | 37 | 51.60% | 543.12 |
| Yang yanfeng [28] | 2022 | NT-ProBNP | 26 | 30 | 19 | 165 | 57.58% | 822 |
| Lv aiting [29] | 2022 | NT-ProBNP | 23 | 8 | 6 | 51 | 79.30% | 587.58 |
| Que xuejun [30] | 2022 | NT-ProBNP | 37 | 6 | 4 | 51 | 90.20% | 797 |
| Sun xinna [31] | 2022 | NT-ProBNP | 26 | 4 | 3 | 47 | 89.20% | 198.4 |
|  |  | CKMB | 25 | 4 | 4 | 47 | 86.50% | —— |
| Jiao lihua [32] | 2023 | NT-ProBNP | 25 | 8 | 22 | 57 | 53.20% | 381.245 |
|  |  | CKMB | 39 | 3 | 8 | 62 | 83.00% | —— |
| Cao hong [33] | 2023 | NT-ProBNP | 15 | 24 | 4 | 56 | 78.90% | 560 |
| Dong nan [34] | 2023 | NT-ProBNP | 9 | 10 | 2 | 43 | 84.78% | —— |
|  |  | CKMB | 9 | 11 | 2 | 42 | 83.59% | —— |
| Liu ya [35] | 2023 | NT-ProBNP | 25 | 10 | 5 | 32 | 86.70% | —— |
|  |  | CKMB | 24 | 9 | 6 | 33 | 83% | —— |

Tp: true positive. Fp:false positive. Fn:fslse negative. Tn:true negative. Sp:specificity. "——": No relevant information is mentioned in the article.

cases. The risk of bias in the included literature was low. The quality evaluation of the included literature is shown in Fig 2 and Fig 3.

### 3.3  Meta analysis results

**3.3.1  Evaluation of NT-ProBNP for cardiac injury in Kawasaki disease.**  There was significant heterogeneity in the combined sensitivity (I2 = 65.45%, P < 0.001), with a random effects estimate of 0.80 [95% CI (0.74, 0.85)]. as illustrated in Fig 4.

Similarly, there was substantial heterogeneity in specificity (I2 = 71.28%, P < 0.001), yielding a result of 0.78 [95% CI (0.75, 0.82)]. as illustrated in Fig 5.

The combined positive likelihood ratio was 3.94 [95% CI (3.32, 4.69)], while the negative likelihood ratio was 0.23 [95% CI (0.17, 0.30)]. The diagnostic odds ratio was 17.24 [95% CI (12.0, 24.77)]. as illustrated in Fig 6.

The area under the SROC curve was AUC = 0.87 (0.84–0.90), as illustrated in Figs 4–7, indicating that NT-ProBNP has high diagnostic value for coronary artery lesions in Kawasaki disease.as illustrated in Fig7

 

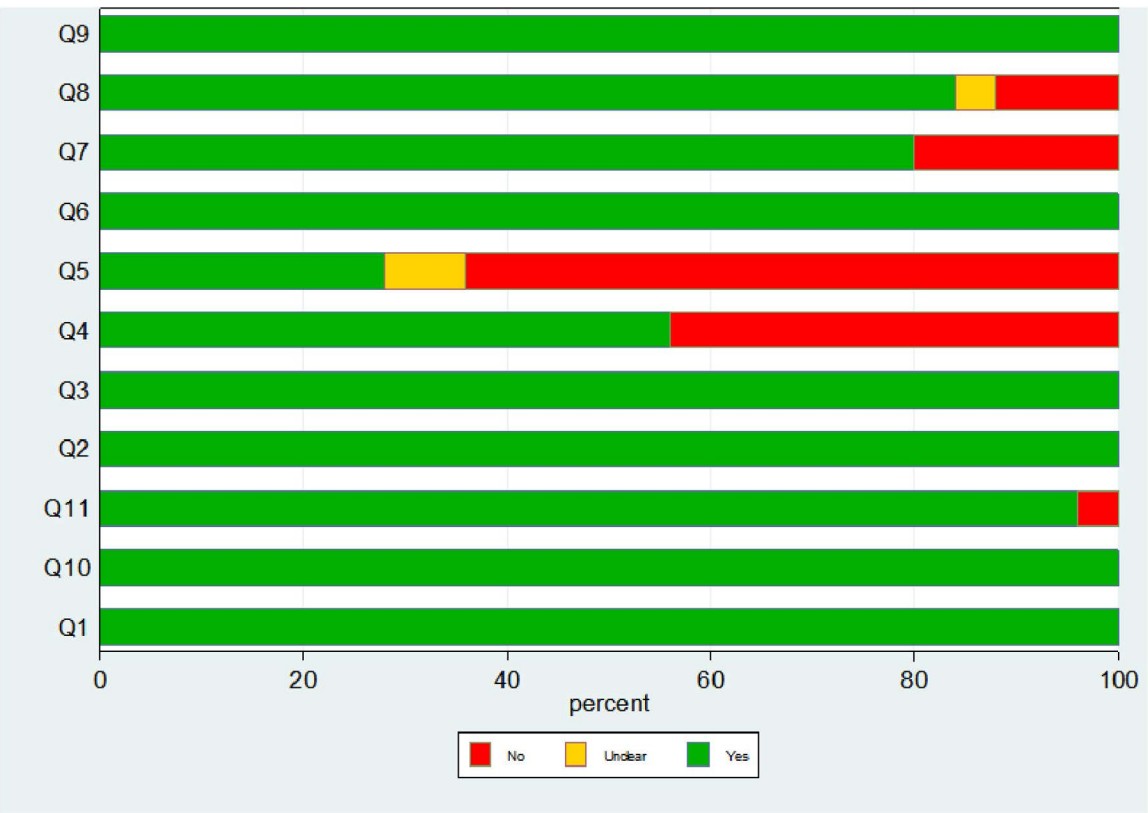

**Fig 2. Literature quality evaluation.** Summary bias risk of included articles using the Newcastle-Ottawa Quality Assessment Scale model.

**3.3.2 Subgroup analysis and predicted value results. Subgroup analysis:** The studies included in the research were divided into a group of less than 3 years old (15 articles) and a group of more than 3 years old (5 articles) based on the average age. It was found that when NT-ProBNP was used to assess cardiac injury in Kawasaki disease, the positive likelihood ratio and diagnostic odds ratio in the group over 3 years old were higher than those in the group under 3 years old (6.4 [4.5, 9.1] vs. 3.6 [3.0, 4.3]) (15 [10,24] vs. 32 [12, 81]), indicating that its diagnostic value and diagnostic efficacy were greater for the group over 3 years old. In the study country group, the sensitivity of NT-ProBNP in evaluating cardiac injury in Kawasaki disease was higher in the Japanese population (0.86) compared to the Chinese population (0.83). However, the specificity was lower in the Japanese population (0.76) than in the Chinese population (0.81). The positive likelihood ratio and negative likelihood ratio for the Japanese population were 4.3 and 0.22, respectively, which were lower than those for the Chinese population (3.6 and 0.18). This suggests that while NT-ProBNP has a higher diagnostic value for the Chinese population, its ability to rule out cardiac injury is less effective compared to the Japanese population. The chemiluminescence method demonstrated superior diagnostic performance compared to ELISA, as evidenced by higher specificity (0.86 vs. 0.79), positive likelihood ratio (5.8 vs. 4.1), and diagnostic odds ratio (31 vs. 20). Additionally, the area under the AUC curve for chemiluminescence was greater (0.91 vs. 0.89). These findings indicate that the chemiluminescence method holds greater diagnostic value.as shown in Table 3.

**Predicted value results:** In the included literature, 15 articles with predicted value <1000 ng/L and 6 articles with 1000 ng/L were combined, which found that when the predicted value was 1000 ng/L, the sensitivity, positive likelihood ratio, diagnostic odds ratio and area under AUC curve of ≥1000 ng/L were higher than <1000 ng/L, indicating that the diagnostic value was >1000 ng/L, and there was no heterogeneity. See Table 4.

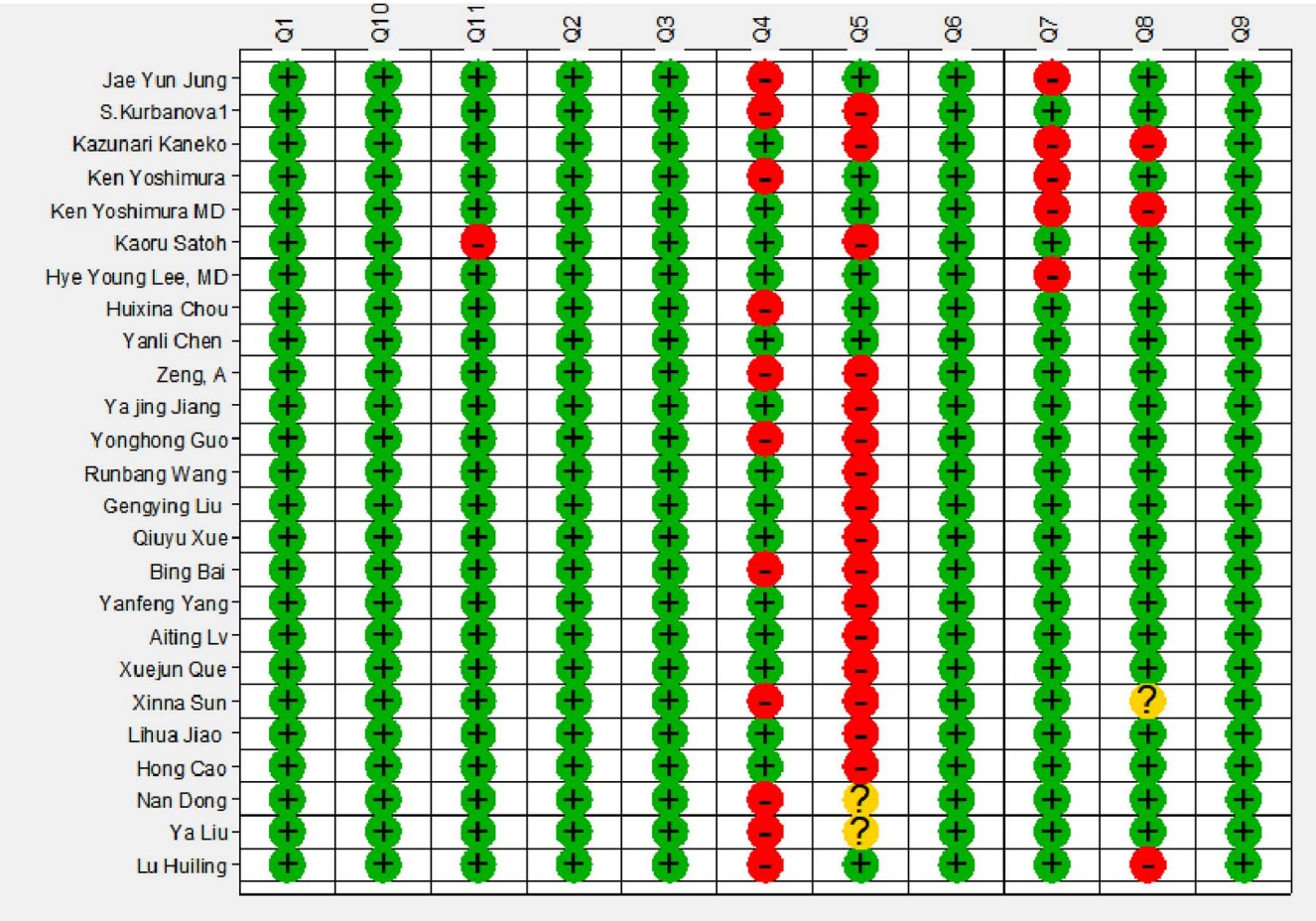

**Fig 3. cMethodological quality of included articles using Newcastle-Ottawa Quality Assessment Scale.**

### 3.4 Bias test results

Diagnostic Meta-analysis heterogeneity originates from publication bias, covariate effects, etc. P = 0.0.92 using the Deep's method (Fig 8) indicates that there was no publication bias between the included studies. Meta regression analysis was conducted using publication time, research methods, research country, research sample size, and research age as covariates in the literature(Fig 9). It was found that the detection method had a significant impact on heterogeneity (I2 = 99, P = 0), research age had a significant impact on heterogeneity (I2 = 95, P = 0), research country had a small impact on heterogeneity (I2 = 63, P = 0.07), publication time had a small impact on heterogeneity (I2 = 21, P = 0.28), and research sample size had no impact on heterogeneity (I2 = 0, P = 0.8). Sensitivity analysis by using the Stata 14.0 software (Fig 10), To assess the impact of a single study on this Meta-analysis, Found that Bai Bing [24], Yang Yanfeng [25], Jiao Lihua [29]3 literature deviated from the confidence interval, After deleting these 3 articles, Meta-analysis, No heterogeneity in the pooled sensitivity (I2 = 0, P = 0.92), The oled specific heterogeneity was reduced (I2 = 61.46, P = 0), Then, a summary analysis of the rest of the study found that any one article was deleted, The confidence intervals of diagnostic parameters largely overlapped with the original data, This indicates that these three articles have a great impact on the heterogeneity. Draw Fagan diagram (Fig 11) after test probability estimation, with Fagan analysis of NT-ProBNP value in evaluating Kawasaki disease, that is, the test probability of the study gold standard was set to 80%, calculation of NT-ProBNP in

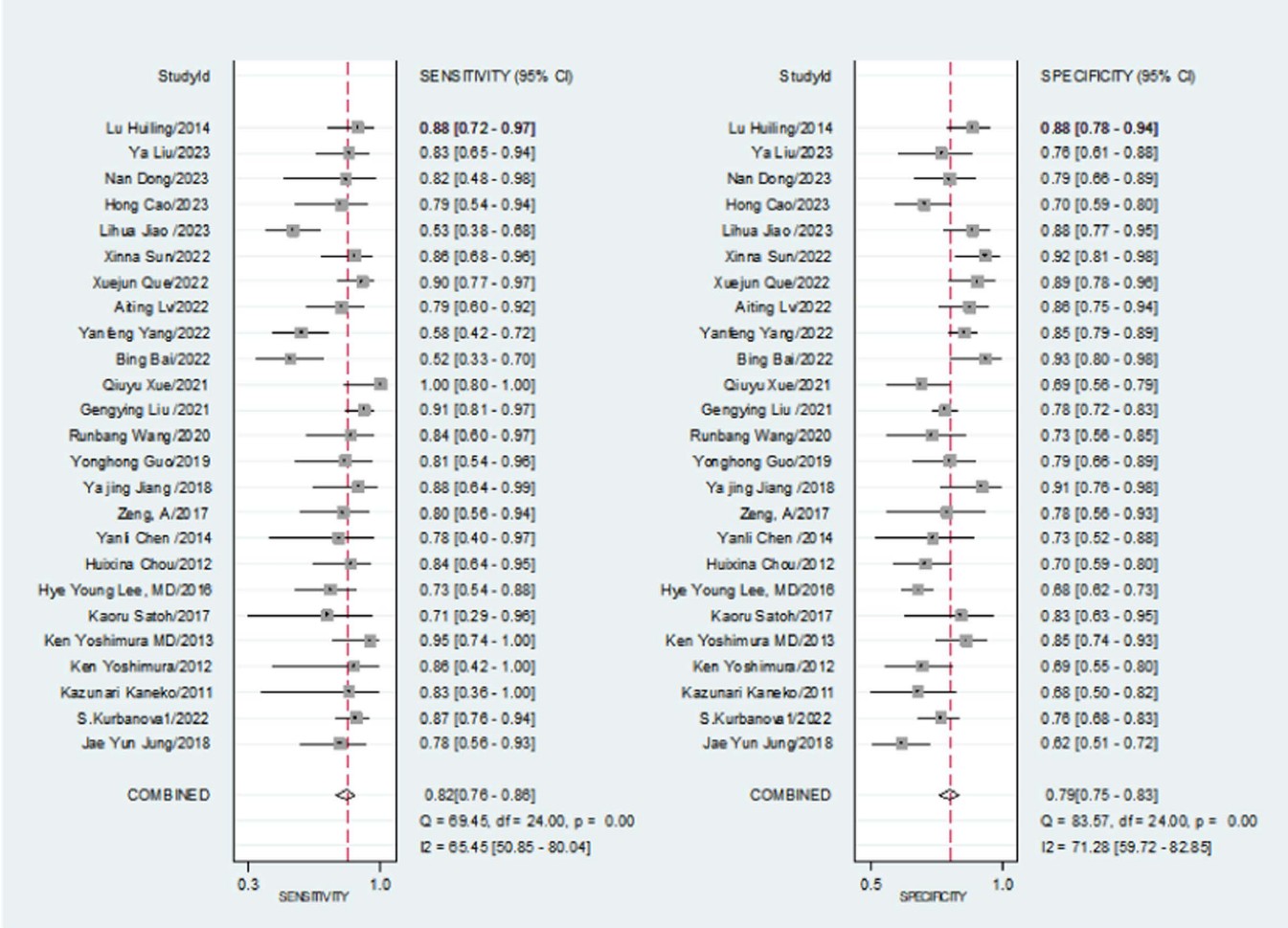

**Fig 4. Meta-analysis of the sensitivity and specificity.**

Kawasaki disease, if the probability of diagnosis of coronary artery lesions is 94%, the exclusion of coronary disease was 44%, indicating that the probability was large, but the exclusion probability was small.

## 4. Discussion

NT-proBNP is a more stable peptide substance secreted by cardiomyocytes, with low blood concentration in normal conditions, and when ventricular wall tension increases or cardiomyocyte ischemia, its secretion increases [36,37]. It is mainly used to diagnose and evaluate the severity of heart failure and its prognosis. In recent years, the diagnosis of cardiac coronary artery lesions in Kawasaki disease has gradually increased. [38]et al showed that NT-proBNP is involved in the pathogenesis of Kawasaki disease and has a close relationship with coronary artery damage. Some studies also found that the elevated level of NT-ProBNP in the acute phase of KD involved [6] in diastotor function, which may be a predictive factor for CAL and IVIg resistance in KD patients [39].Therefore, the results of this Meta analysis with small samples showed that NT-ProBNP had high total Se and Sp (0.80 and 0.79, respectively), which further confirmed the conclusions of the previous study. AUC = 0.87, close to 1, with a high diagnostic value. The total Sp and AUC were higher than the previous study (total Sp 0.71, AUC = 0.85) [10], and the combined total PLR and NLR were 3.97 and 0.23, respectively.

**Fig 5. Meta-analysis of the positive likelihood ratio and the negative relief ratio.**

When PLR > 10 or NLR < 0.1, the possibility of diagnosis or exclusion of a disease was significant. The results of this study do not support the exclusion of coronary artery lesions in Kawasaki disease when NT-ProBNP detection threshold is low. The DOR is 17.24, and the greater the value is, the better the discriminative effect of the diagnostic test is. This indicates that the discriminative effect and accuracy of the results are relatively high.

The findings of this study exhibit considerable inconsistency. The sources of heterogeneity can be examined from three perspectives: covariate effects, publication bias, and sensitivity analysis. Given that the Deep method did not detect publication bias in this study, further sensitivity analysis revealed that the exclusion of three papers [24,26,31] significantly impacted result stability, reducing heterogeneity without causing a significant difference in overall outcomes (Se 0.80 vs. 0.85, Sp 0.79 vs. 0.78). This indicates that variations among the included studies did not substantially affect the results. Meta-regression analysis of relevant covariates showed that sample size had no effect on heterogeneity (I2 = 0, P = 0.91), while publication volume exhibited minimal heterogeneity (I2 = 27, P = 0.26). In contrast, the country of study exerted a substantial influence on heterogeneity (I2 = 63, P = 0.07), consistent with previous systematic reviews [40]. However, strong heterogeneity was observed in age and detection methods.

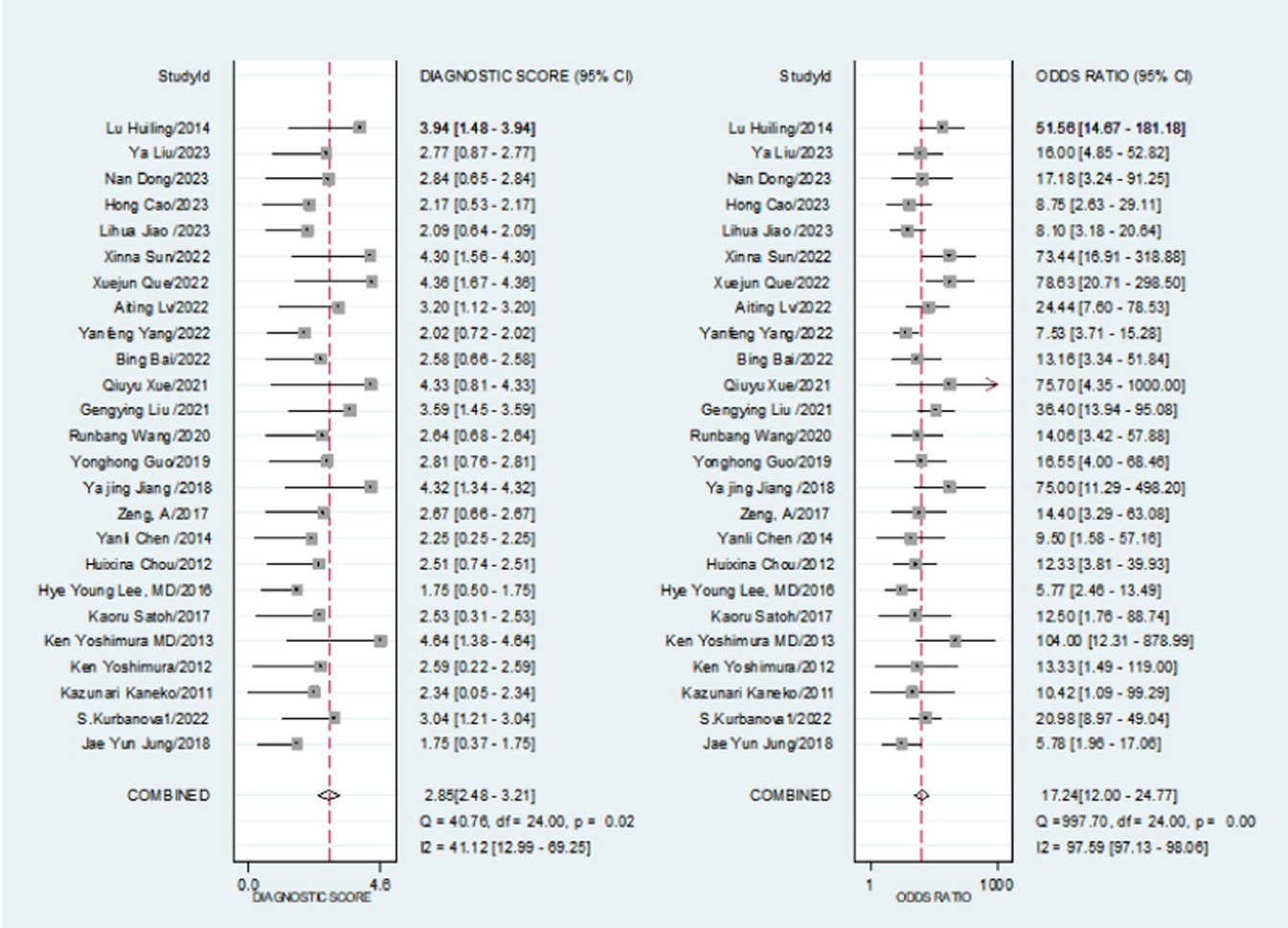

**Fig 6. Meta-analysis of the diagnostic odds ratio.**

Subgroup analysis with age (< 3 years old and ≥3 years old), country (China and Japan), and detection method (chemiluminescence and ELISA) as covariates showed that the diagnostic value of NT-ProBNP combined with CAL was greater in the group older than 3 years old (6.4VS3.6). This may be because NT-proBNP levels are highest in the first few weeks after birth and gradually decrease with age [41,39]. It has also been reported that the level of NT-proBNP in infants (2442±1866 pg/mL) is significantly higher than that in children (945±1151 pg/mL) [42], so a high level of serum NT-proBNP concentration when children are over 3 years old is more valuable for predicting coronary artery injury in Kawasaki disease. The sensitivity and specificity of detection of NT-ProBNP by chemiluminescence method are higher than those by enzyme-linked immunoassay. Liu Ju [43] compared the two detection methods to detect serological markers of hepatitis B virus infection and concluded that chemiluminescence method has better sensitivity than enzyme-linked immunosorbent assay. The study of David L Brandon [44] describes electrochemical luminescence (ECL) detection as using multiple excitation cycles to amplify the luminous signal and improve sensitivity. The excitation mechanism and the relatively long emission wavelength (620 nm) may have the ability to resist matrix effects. Although ECL instruments and reagents are relatively expensive, the coefficient of variation is lower and the detection sensitivity is high. Enzyme-linked immunosorbent assay (ELISA) materials are relatively inexpensive. But the sensitivity is low.The findings suggest that the

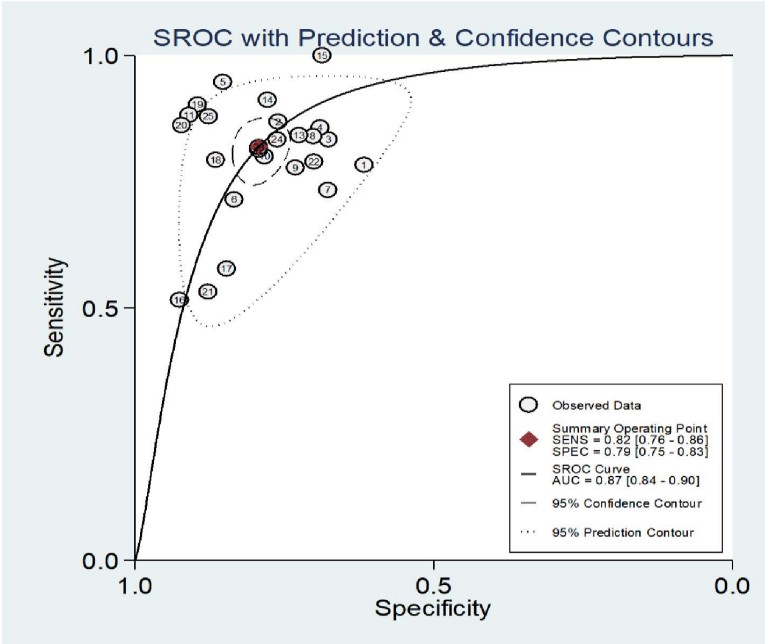

**Fig 7. Area under the SROC curve.**

**Table 3. Results of the subgroup analysis.**

| Group | Documents | Se | Sp | PLR | NLR | DOR | AUC |
|---|---|---|---|---|---|---|---|
| Age | | | | | | | |
| average age < 3year | 15 | 0.82 [0.75,0.87] | 0.77 [0.73,0.81] | 3.6 [3.0,4.4] | 0.24 [0.17,0.33] | 15 [10,24] | 0.86 [0.83,0.89] |
| average age > 3year | 5 | 0.82 [0.66,0.91] | 0.87 [0.82,0.90] | 6.4 [4.5,9.1] | 0.22 [0.11,0.42] | 32 [12,81] | 0.88 [0.85,0.90] |
| country | | | | | | | |
| China | 18 | 0.83 [0.76,0.88] | 0.81 [0.77,0.84] | 4.3 [3.6,5.1] | 0.22 [0.16,0.29] | 20 [1 4,29] | 0.88 [0.85,0.90] |
| Japan | 4 | 0.86 [0.70,0.95] | 0.76 [0.67,0.84] | 3.6 [2.4,5.5] | 0.18 [0.07,0.44] | 20 [6,68] | 0.89 [0.85,0.91] |
| test method | | | | | | | |
| Chemiluminescence method | 7 | 0.84 [0.71,0.92] | 0.86 [0.80,0.90] | 5.8 [4.0,8.5] | 0.19 [0.10,0.36] | 31 [13,76] | 0.91 [0.88,0.93] |
| Immunoassay method | 5 | 0.84 [0.76,0.90] | 0.79 [0.73,0.85] | 4.1 [3.0,5.5] | 0.20 [0.12,0.32] | 20 [10,36] | 0.89 [0.86,0.91] |

Se:sensitivity. Sp:specificity. PLR:positive likelihood ratio. NLR: negative likelihood ratio. DOR:diagnostic odds ratio.AUC:area under curve.

**Table 4. Results of the predictive value analysis.**

| Group | Documents | Se | Sp | NLR | RLR | DOR | AUC | Se | | Sp | |
|---|---|---|---|---|---|---|---|---|---|---|---|
| | | | | | | | | p | I² | p | I² |
| Estimate < 1000ng/L | 17 | 0.80 [0.73,0.86] | 0.80 [0.75,0.84] | 3.95 | 0.25 | 16 | 0.87 | 0 | 71.98 | 0 | 78.04 |
| Estimate > 1000ng/L | 6 | 0.87 [0.80,0.92] | 0.78 [0.71,0.84] | 4.03 | 0.16 | 24 | 0.90 | 0.75 | 0 | 0.05 | 55.22 |

Se:sensitivity. Sp:specificity. PLR:positive likelihood ratio. NLR: negative likelihood ratio. DOR:diagnostic odds ratio.AUC:area under curve.

 

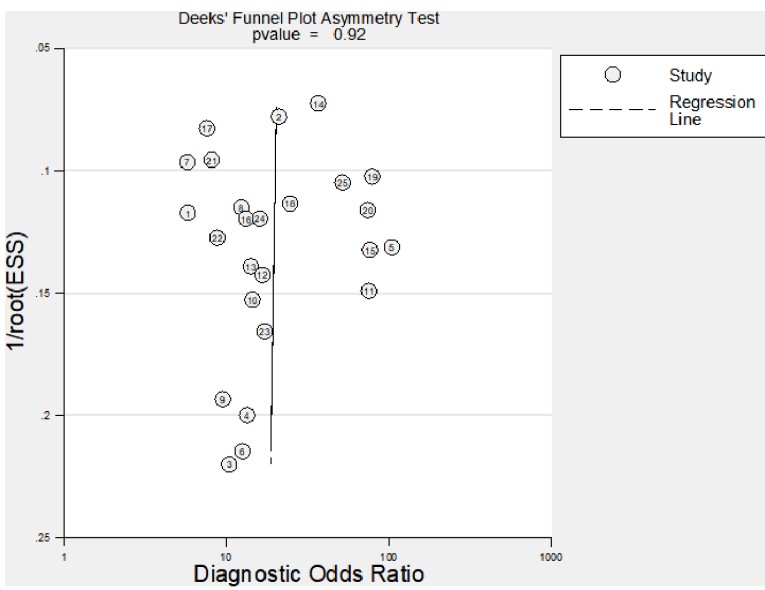

**Fig 8. Deep's Fig.**

| Parameter | category | LRTChi2 | Pvalue | I2 | I2lo | I2hi |
|---|---|---|---|---|---|---|
| studycountry | Yes | 5.35 | 0.07 | 63 | 16 | 100 |
|  | No | . | . | . | . | . |
| publicationtime | Yes | 2.55 | 0.28 | 21 | 0 | 100 |
|  | No | . | . | . | . | . |
| studyage | Yes | 36.88 | 0.00 | 95 | 90 | 99 |
|  | No | . | . | . | . | . |
| researchmethod | Yes | 158.77 | 0.00 | 99 | 98 | 99 |
|  | No | . | . | . | . | . |
| samplesize | Yes | 0.42 | 0.81 | 0 | 0 | 100 |
|  | No | . | . | . | . | . |

**Fig 9. Meta regression analysis.**

accuracy of chemiluminescence detection is superior; however, variations in the manufacturers or batches of kits utilized in the study may introduce inconsistencies, necessitating further investigation.In the national subgroup analysis, the Japanese population exhibited higher sensitivity compared to the Chinese population; however, the specificity, positive likelihood ratio, and negative likelihood ratio were lower in the Japanese population. These findings suggest that NT-ProBNP holds greater diagnostic value for the Chinese population and a higher exclusion value for the Japanese population when evaluating coronary artery lesions in Kawasaki disease. A comprehensive literature review revealed no direct comparisons between Chinese and Japanese populations, which may be attributed to differences in detection methods between the two countries. Specifically, two out of four studies in Japan utilized chemiluminescence, whereas only five out of eighteen studies in China explicitly employed this method. This discrepancy could be related to the difference in sensitivity between chemiluminescence and ELISA. Additionally, variations in population constitution and reagents used by different manufacturers may also contribute to these differences, warranting further investigation. The results are shown in Table 3.

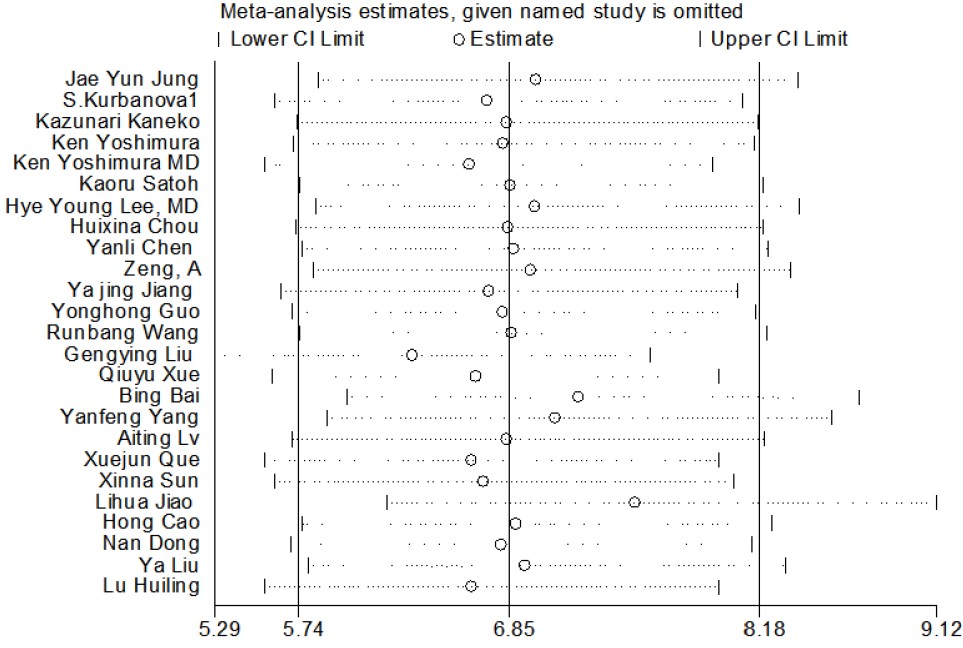

**Fig 10. Sensitivity analysis.**

In this study, the predicted values were divided into <1000 ng/L and ≥1000 ng/L groups, and the analysis found that the Se and Sp were slightly above the overall diagnostic accuracy of 1000 ng/L. However, the Se and Sp and overall diagnostic accuracy of NT-ProBNP were <1000 ng/L. This indicates that the higher the diagnostic threshold, the greater the accuracy of NT-ProBNP for early assessment of coronary lesions in Kawasaki disease. Several studies have indicated that an NT-proBNP level exceeding 700 pg/mL in febrile patients should raise a high suspicion of Kawasaki disease [45]. Furthermore, one study proposed that a serum NT-proBNP concentration greater than 1,000 pg/mL strongly predicts the presence of coronary artery disease, which is consistent with the findings of this study [44]. Consequently, when NT-proBNP levels exceed 1,000 pg/mL in pediatric patients with Kawasaki disease, heightened vigilance for potential coronary artery damage and enhanced follow-up measures are warranted. Nonetheless, this study has certain limitations, as some investigations failed to establish a threshold for NT-proBNP levels, leading to significant biases in the execution or interpretation of the experiments under evaluation.The recommended threshold still needs to be further determined by more well-designed studies with larger sample size, shown in Table 4.

Although the sample size is large in this study, there are still limitations: 1. The included children are Asian populations and can not be widely used; 2. Although 4 articles simultaneously studied CK-MB, there is still no research data combining the two; 3. The diagnostic threshold of 16 of the included 25 studies was not determined before the experiment, which may cause deviation to the experimental results, and the diagnostic threshold can not be determined; 4. the included articles are published literature, and some grey research results may be missed, and relevant literature except in Chinese and English.

Despite the aforementioned limitations, this study encompasses a substantial sample size, enhancing its credibility. The majority of publications included are from post-2020, aligning closely with the evolving characteristics of the disease. This paper further examines age differences and detection methods between Japan and China, evaluating predicted values based on an original systematic review. In clinical practice, NT-ProBNP can be measured via chemiluminescence

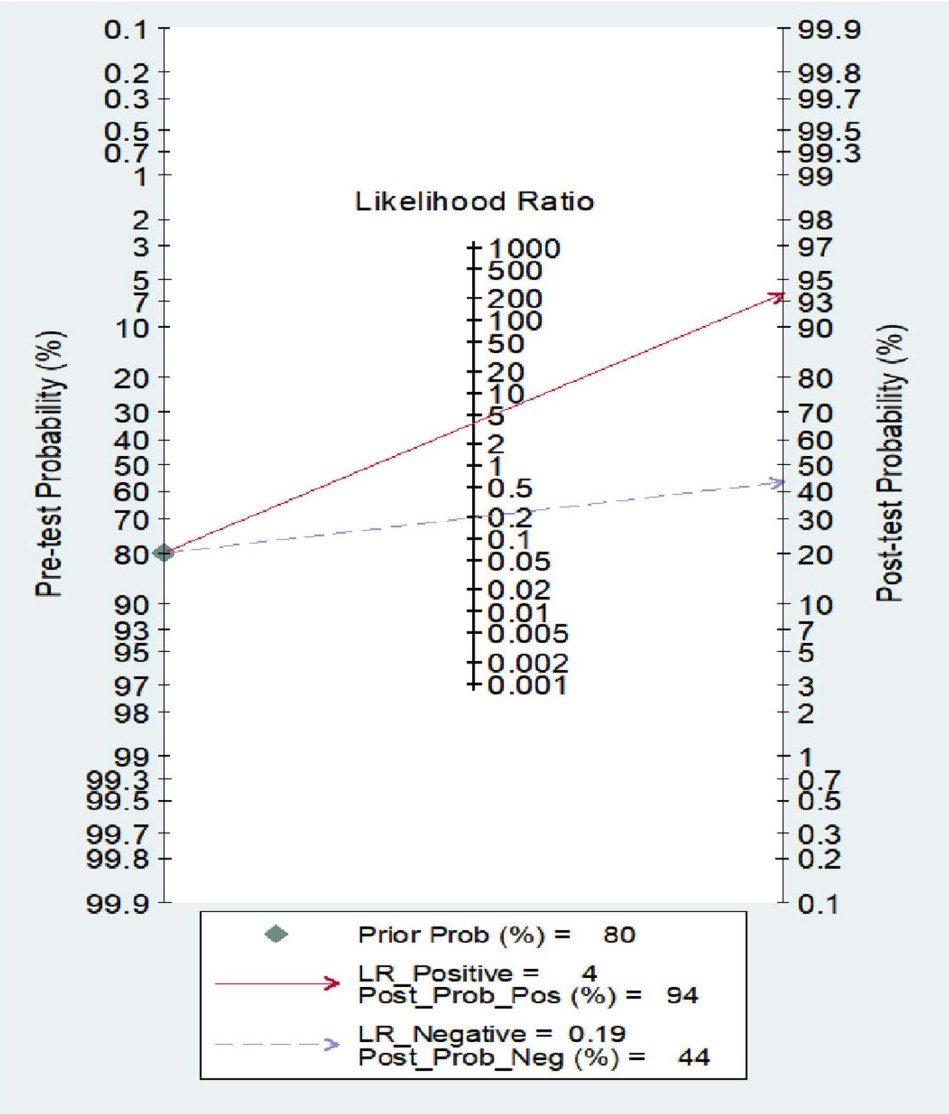

**Fig 11. Fagan analysis.**

in children over three years old diagnosed with Kawasaki disease, demonstrating significant value in assessing coronary artery damage. An NT-ProBNP level ≥1000ng/L effectively predicts coronary artery damage. NT-ProBNP exhibits greater diagnostic value for predicting coronary artery damage in the Chinese population and better exclusion value in the Japanese population. However, the diagnostic efficacy of NT-ProBNP requires validation through larger sample sizes and more precise study designs.

## 5. Conclusion

NT-ProBNP has high diagnostic value for coronary artery lesions in Kawasaki disease, and has greater diagnostic value for children over 3 years old, and the use of chemiluminescence.It demonstrates superior diagnostic efficacy for the Chinese population and enhanced exclusionary power for the Japanese population.

6. AcknowledgmentsWe thank our colleagues for missing full-text articles and their invaluable help to translate the articles. The authors received no specific funding for this work and declare that no competing interests exist.

## Author contributions

**Conceptualization:** Yanyan Li, Chaolong Zheng.

**Data Curation:** Yanyan Li, Chaolong Zheng.

**Formal analysis:** Chaolong Zheng, Sisi Cheng.

**Funding acquisition:** Sisi Cheng.

**Investigation:** Sisi Cheng.

**Methodology:** Sisi Cheng, Limin Chu, Guiling Liu.

**Project administration:** Limin Chu, Guiling Liu.

**Resources:** Limin Chu, Zhiqing Chen, Guiling Liu.

**Software:** Limin Chu, Zhiqing Chen, Guiling Liu.

**Supervision:** Zhiqing Chen, Guiling Liu.

**Validation:** Yanyan Li, Guiling Liu.

**Visualization:** Yanyan Li, Guiling Liu.

**Writing – original draft:** Yanyan Li, Chaolong Zheng.

**Writing – review & editing:** Yanyan Li, Chaolong Zheng.

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
