## [Decision Letter · Decision Letter 0]

PONE-D-24-44564Value of the latest N-terminal brain natriuretic peptide precursors in predicting coronary artery injury in Kawasaki disease: A meta-analysisPLOS ONE

Dear Dr. yan yan,

Thank you for submitting your manuscript to PLOS ONE. After careful consideration, we feel that it has merit but does not fully meet PLOS ONE’s publication criteria as it currently stands. Therefore, we invite you to submit a revised version of the manuscript that addresses the points raised during the review process.

**ACADEMIC EDITOR: **

- Please provide detailed response for the each of the Reviewers' comments as specified below.

- Please elaborate on the regional differences in the diagnostic accuracy of NTproBNP. What are you possible explanation for such findings.

- Based on the results of your study, please provide guidance for clinicians on how to use and interpret NPproBNP levels in clinical practice depending on the patient's age and other demographics. You can also make a figure to better illustrate it. Also, please explain clinical utility of your findings.

We look forward to receiving your revised manuscript.

Kind regards,

Milos Brankovic, MD, PhD, MSc

Academic Editor

PLOS ONE

Journal Requirements:

1. Please ensure that your manuscript meets PLOS ONE's style requirements, including those for file naming. The PLOS ONE style templates can be found at https://journals.plos.org/plosone/s/file?id=wjVg/PLOSOne_formatting_sample_main_body.pdf and https://journals.plos.org/plosone/s/file?id=ba62/PLOSOne_formatting_sample_title_authors_affiliations.pdf.

2. Please update your submission to use the PLOS LaTeX template. The template and more information on our requirements for LaTeX submissions can be found at http://journals.plos.org/plosone/s/latex .

3. As required by our policy on Data Availability, please ensure your manuscript or supplementary information includes the following:

6. Please remove your figures from within your manuscript file, leaving only the individual TIFF/EPS image files, uploaded separately. These will be automatically included in the reviewers’ PDF.

Reviewers' comments:

Reviewer's Responses to Questions

**Comments to the Author**

1. Is the manuscript technically sound, and do the data support the conclusions?

Reviewer #1: Yes

Reviewer #2: Partly

Reviewer #3: Yes

2. Has the statistical analysis been performed appropriately and rigorously? 

Reviewer #1: I Don't Know

Reviewer #2: I Don't Know

Reviewer #3: Yes

3. Have the authors made all data underlying the findings in their manuscript fully available?

Reviewer #1: Yes

Reviewer #2: No

Reviewer #3: Yes

4. Is the manuscript presented in an intelligible fashion and written in standard English?

Reviewer #1: Yes

Reviewer #2: Yes

Reviewer #3: Yes

5. Review Comments to the Author

Reviewer #1: Congratulations on a thoughtful review of recent literature concerning BNP as a predictor of coronary artery lesions in Kawasaki disease. I think this is a useful addition to the existing literature. My one recommendation is to provide a simple clinician focused summary statement (in the abstract and in the conclusions) about how clinicians should use NT-Pro-BNP when evaluating a child with known or suspected kawasaki disease. For example if that child's Level is >1000, what does your meta-analysis suggest to that clinician? Should BNP be routinely checked in all children with Kawasaki disease?

Reviewer #2: 1. General comments

This study investigates the diagnostic value of N-terminal pro-brain natriuretic peptide (NT-proBNP) for coronary artery lesions (CAL) in Kawasaki disease (KD) through a meta-analysis of 23 studies with 2,218 patients. NT-proBNP showed high diagnostic accuracy with sensitivity and specificity of 0.80 and 0.79, respectively, and an area under the curve (AUC) of 0.87. Subgroup analysis highlighted superior diagnostic efficacy for children over three years old and the Japanese population. Chemiluminescence assays outperformed enzyme-linked immunoassays in accuracy. NT-proBNP levels above 1000 ng/L demonstrated improved sensitivity and diagnostic odds ratio. Despite heterogeneity, these results suggested NT-proBNP as a reliable biomarker for early CAL detection in KD. Limitations include focus on Asian populations and varied study thresholds, necessitating further validation through larger, diverse cohorts. Findings provide vital evidence for enhancing early CAL diagnosis and management in KD patients.

Numerous studies have been conducted on NT-proBNP in KD, and this study is intriguing from the perspective of reevaluating its value. However, there are several major concerns.

2. Specific comments

P2: You described “Although coronary angiography and cardiac ultrasound are considered gold standards for diagnosing CAL, these methods have inherent limitations and lack sufficient sensitivity.” The term "inherent limitations" is ambiguous. It should be more clearly specified what aspects of coronary artery ultrasound and coronary angiography are problematic, and why improving the predictive accuracy for CAL using biomarkers such as NT-proBNP would be preferable. However, in clinical settings, if coronary artery ultrasound is performed correctly, it is often possible to diagnose CAL without relying on NT-proBNP.

P3 1.1: Due to the nature of the study design, a Chinese-specific literature database has been used. However, could there be references included that are not accessible in English? If the references are only available in Chinese, reviewers will not be able to evaluate their content.

P3 1.2: In this study, although the evaluation of whether sufficient information regarding NT-proBNP was provided in patient selection appears to be adequate, the examination of the definition of KD, the definition of CAL, and the methods used to assess CAL is insufficient. This is one of the major concerns of the study.

P9: Although you mentioned that NT-proBNP had higher diagnostic value in older children, could it not be argued that younger children have narrower confidence intervals, suggesting higher test accuracy?

Reviewer #3: I have carefully reviewed your manuscript, titled "Value of the latest N-terminal brain natriuretic peptide precursors in predicting coronary artery injury in Kawasaki disease: A meta-analysis." This study provides a significant contribution by systematically evaluating the relationship between NT-proBNP and Kawasaki disease, particularly its diagnostic value for coronary artery lesions.

However, major and minor revisions of manuscript is needed before it can be accepted for publication.

1. The reference value of NT-proBNP is significantly higher in infants under 1 year of age, and this age group should be considered separately. In particular, the level of NT-proBNP is very high in infants under 6 months. It is true that this period is associated with a high incidence of coronary artery lesions (CALs), but it is not directly correlated with elevated NT-proBNP levels. Please analyze and consider this point.

2. In Table 1, age should be listed as the median [quartile] instead of the average.

3. In the “test method” in Table 3, I think the second row is not the “chemiluminescence method”, but “ELISA”

6. PLOS authors have the option to publish the peer review history of their article (what does this mean? ). If published, this will include your full peer review and any attached files.

**Do you want your identity to be public for this peer review?** For information about this choice, including consent withdrawal, please see our Privacy Policy .

Reviewer #1: No

Reviewer #2: No

Reviewer #3: No

---

## [Author Response · Author response to Decision Letter 1]

3 Feb 2025

1、Description of the inherent limitations of coronary angiography on coronary artery injury

Although echocardiography and coronary angiography are the gold standard for diagnosing coronary artery injury, many young children are difficult to soothe because they need complete quiet cooperation for echocardiography. Coronary angiography diagnosis requires interventional operation, which has great side effects and high cost for children, and many families of children have concerns . At present, there is still a lack of early simple and accurate assessment of CAL in children with Kawasaki disease. Therefore, it is of great clinical significance to find indicators for early diagnosis of KD and prevention of CAL [5].

2、Please elaborate on the regional differences in the diagnostic accuracy of NTproBNP. What are you possible explanation for such findings.Based on the results of your study, please provide guidance for clinicians on how to use and interpret NPproBNP levels in clinical practice depending on the patient's age and other demographics. You can also make a figure to better illustrate it. 

Subgroup analysis with age (< 3 years old and ≥3 years old), country (China and Japan), and detection method (chemiluminescence and ELISA) as covariates showed that the diagnostic value of NT-ProBNP combined with CAL was greater in the group older than 3 years old (6.4VS3.6). This may be because NT-proBNP levels are highest in the first few weeks after birth and gradually decrease with age . It has also been reported that the level of NT-proBNP in infants (2442±1866 pg/mL) is significantly higher than that in children (945±1151 pg/mL) , so a high level of serum NT-proBNP concentration when children are over 3 years old is more valuable for predicting coronary artery injury in Kawasaki disease. The sensitivity and specificity of detection of NT-ProBNP by chemiluminescence method are higher than those by enzyme-linked immunoassay. Liu Ju [42] compared the two detection methods to detect serological markers of hepatitis B virus infection and concluded that chemiluminescence method has better sensitivity than enzyme-linked immunosorbent assay. The study of David L Brandon[43] describes electrochemical luminescence (ECL) detection as using multiple excitation cycles to amplify the luminous signal and improve sensitivity. The excitation mechanism and the relatively long emission wavelength (620 nm) may have the ability to resist matrix effects. Although ECL instruments and reagents are relatively expensive, the coefficient of variation is lower and the detection sensitivity is high. Enzyme-linked immunosorbent assay (ELISA) materials are relatively inexpensive. But the sensitivity is low.The findings suggest that the accuracy of chemiluminescence detection is superior; however, variations in the manufacturers or batches of kits utilized in the study may introduce inconsistencies, necessitating further investigation. In the national subgroup analysis, the sensitivity, diagnostic odds ratio, and area under the AUC curve for the Japanese population were consistently higher than those for the Chinese population, indicating that NT-ProBNP has greater diagnostic utility for the Japanese population. A comprehensive search of the literature revealed no studies directly comparing the Chinese and Japanese populations, and the observed differences may be attributed to variations in detection methodologies between the two countries. Among the four Japanese studies included, two employed the chemiluminescence method, whereas among the seventeen Chinese studies, only five definitively used this method, suggesting that the enhanced sensitivity of the chemiluminescence method compared to the ELISA method could contribute to these discrepancies.However, it may also be related to the physical fitness of the population and the different reagents used in the detection of different manufacturers, and the results need to be further discussed. The results are shown in Table 3.

In this study, the predicted values were divided into <1000 ng/L and 1000 ng/L groups, and the analysis found that the Se and Sp were slightly above the overall diagnostic accuracy of 1000 ng/L. However, the Se and Sp and overall diagnostic accuracy of NT-ProBNP were <1000 ng/L. This indicates that the higher the diagnostic threshold, the greater the accuracy of NT-ProBNP for early assessment of coronary lesions in Kawasaki disease. Several studies have indicated that an NT-proBNP level exceeding 700 pg/mL in febrile patients should raise a high suspicion of Kawasaki disease. Furthermore, one study proposed that a serum NT-proBNP concentration greater than 1,000 pg/mL strongly predicts the presence of coronary artery disease, which is consistent with the findings of this study . Consequently, when NT-proBNP levels exceed 1,000 pg/mL in pediatric patients with Kawasaki disease, heightened vigilance for potential coronary artery damage and enhanced follow-up measures are warranted. Nonetheless, this study has certain limitations, as some investigations failed to establish a threshold for NT-proBNP levels, leading to significant biases in the execution or interpretation of the experiments under evaluation.The recommended threshold still needs to be further determined by more well-designed studies with larger sample size, shown in Table 4.

4、Clinician's advice

Although this paper has the above limitations, the sample size included is large and more convincing, and published after 2020, which is more in line with the characteristics of disease development.Furthermore, the study analyzed the differences in age, between Japan and China, as well as the detection method, and evaluated the prediction values based on the original system evaluation. In clinical practice, children with Kawasaki disease who are over 3 years old can be tested for NT-ProBNP using a chemiluminescence method. An NT-ProBNP level of ≥1000ng/L can strongly predict coronary damage, and NT-ProBNP is more sensitive in predicting coronary damage in the Japanese population. To provide more precise clinical evidence for the KD-CAL fusion. However, a larger sample size and a more precise research design are needed to determine the diagnostic value of NT-ProBNP.

---

## [Decision Letter · Decision Letter 1]

PONE-D-24-44564R1Value of the latest N-terminal brain natriuretic peptide precursors in predicting coronary artery injury in Kawasaki disease: A meta-analysisPLOS ONE

Dear Dr. yan yan,

Thank you for submitting your manuscript to PLOS ONE. After careful consideration, we feel that it has merit but does not fully meet PLOS ONE’s publication criteria as it currently stands. Therefore, we invite you to submit a revised version of the manuscript that addresses the points raised during the review process.

- Included studies are predominantly in non-English language which limits the broad readership to assess the references. Could authors analyze the effect of whether the studies were published in English or non-English language.

- Could authors further define if the subgroup analysis is >3yrs and 1-3 yrs or infants were also included in the study. Please see Reviewer's prior comment on different study population, specifically infants.

 - Please make sure that revised tables and figures reflect the changes that are noted in the text.

- We suggest authors to review the manuscript for typographical and grammatical errors this will make correlating the results with the proposed conclusions more understandable and improve the overall readability of the paper.

We look forward to receiving your revised manuscript.

Kind regards,

Milos Brankovic, MD, PhD, MSc

Academic Editor

PLOS ONE

Journal Requirements:

Additional Editor Comments (if provided):

- Included studies are predominantly in non-English language which limits the broad readership to assess the references. Could authors analyze the effect of whether the studies were published in English or non-English language.

- Could authors further define if the subgroup analysis is >3yrs and 1-3 yrs or infants were also included in the study. Please see Reviewer's prior comment on different study population, specifically infants.

 - Please make sure that revised tables and figures reflect the changes that are noted in the text.

- We suggest authors to review the manuscript for typographical and grammatical errors this will make correlating the results with the proposed conclusions more understandable and improve the overall readability of the paper.

Reviewers' comments:

Reviewer's Responses to Questions

**Comments to the Author**

1. If the authors have adequately addressed your comments raised in a previous round of review and you feel that this manuscript is now acceptable for publication, you may indicate that here to bypass the “Comments to the Author” section, enter your conflict of interest statement in the “Confidential to Editor” section, and submit your "Accept" recommendation.

Reviewer #1: All comments have been addressed

Reviewer #2: (No Response)

Reviewer #3: All comments have been addressed

2. Is the manuscript technically sound, and do the data support the conclusions?

Reviewer #1: Partly

Reviewer #2: No

Reviewer #3: Yes

3. Has the statistical analysis been performed appropriately and rigorously? 

Reviewer #1: I Don't Know

Reviewer #2: No

Reviewer #3: Yes

4. Have the authors made all data underlying the findings in their manuscript fully available?

Reviewer #1: Yes

Reviewer #2: No

Reviewer #3: Yes

5. Is the manuscript presented in an intelligible fashion and written in standard English?

Reviewer #1: No

Reviewer #2: No

Reviewer #3: Yes

6. Review Comments to the Author

Reviewer #1: I congratulate the authors on a diligent attempt to synthesize a heterogenous data sample. I continue to think this work has the potential to answer an important question regarding the role of BNP in risk stratifying KD patients. The clinical relevance is summarized nicely, albeit limited by the data quality, in the conclusion. I encourage the authors to review the manuscript for typographical and grammatical errors this will make correlating the results with the proposed conclusions more understandable and improve the overall readability of the paper.

Reviewer #2: The authors have not sufficiently addressed the reviewers' questions in their responses. While some points may have been acknowledged, the explanations provided are not comprehensive enough to fully satisfy the concerns raised by the reviewers.

Reviewer #3: Authors have adequately addressed my comments. The review methodology is adequate and the data support the conclusions.

7. PLOS authors have the option to publish the peer review history of their article (what does this mean? ). If published, this will include your full peer review and any attached files.

**Do you want your identity to be public for this peer review?** For information about this choice, including consent withdrawal, please see our Privacy Policy .

Reviewer #1: No

Reviewer #2: No

Reviewer #3: No

---

## [Author Response · Author response to Decision Letter 2]

20 Apr 2025

Dear editor, I re-searched CNKI, Wanfang and Wipu data using Kawasaki disease, cutaneous mucous lymph node syndrome, coronary artery injury, infants, and midbrain natriuretic peptide precursors as keywords in order to find relevant literature. In addition, we searched Kawasaki disease, infants, coronary artery injury, and N-terminal natriuretic peptide precursors in Pubmed and embase databases, but no related literature was found. The median age of all children in the literature included in this article was over 1 year, so it is particularly unfortunate that the study could not be conducted at 1 year or 6 months of age.

Subgroup analysis with age (< 3 years old and ≥3 years old), as covariates showed that the diagnostic value of NT-ProBNP combined with CAL was greater in the group older than 3 years old (6.4VS3.6). This may be because NT-proBNP levels are highest in the first few weeks after birth and gradually decrease with age . It has also been reported that the level of NT-proBNP in infants (2442±1866 pg/mL) is significantly higher than that in children (945±1151 pg/mL) , so a high level of serum NT-proBNP concentration when children are over 3 years old is more valuable for predicting coronary artery injury in Kawasaki disease.

---

## [Editor Report · Decision Letter 2]

Value of the latest N-terminal brain natriuretic peptide precursors in predicting coronary artery injury in Kawasaki disease: A meta-analysis

PONE-D-24-44564R2

Dear Dr. yan yan,

We’re pleased to inform you that your manuscript has been judged scientifically suitable for publication and will be formally accepted for publication once it meets all outstanding technical requirements.

Kind regards,

Milos Brankovic, MD, PhD, MSc

Academic Editor

PLOS ONE
---

## [Editor Report · Acceptance letter]

PONE-D-24-44564R2

PLOS ONE

Dear Dr. yan yan,

I'm pleased to inform you that your manuscript has been deemed suitable for publication in PLOS ONE. Congratulations! Your manuscript is now being handed over to our production team.

Kind regards,

on behalf of

Dr. Milos Brankovic

Academic Editor

PLOS ONE